# The Effect of Induction Chemotherapy with VEGF Inhibition on Tumor Response in Synchronously Metastasized Potentially Resectable Colorectal Cancer

**DOI:** 10.3390/cancers15112900

**Published:** 2023-05-24

**Authors:** Rebecca Thonhauser, Marcus Poglitsch, Jan Philipp Jonas, Yawen Dong, Madita Tschögl, Mariel Gramberger, Mohamed Salem, Jonas Santol, Irmgard Brandl, Martin Klimpfinger, Constantin Vierziger, Thomas Gruenberger

**Affiliations:** 1Department of Surgery, HPB Center, Clinic Favoriten, 1100 Vienna, Austriatellsantol@gmail.com (J.S.); 2Institute of Pathology and Bacteriology, Clinic Favoriten, 1100 Vienna, Austria; 3Institute for Diagnostic and Interventional Radiology, Clinic Favoriten, 1100 Vienna, Austria

**Keywords:** colorectal cancer, synchronous liver metastases, targeted therapy, tumor regression grading

## Abstract

**Simple Summary:**

Synchronously metastasized colorectal cancer (mCRC) is a disease with high morbidity and mortality; therefore, urgent therapeutic decisions are necessary. In recent years, the therapeutic strategies and outcomes have improved due to systemic and local therapies. The primary aim of this retrospective study was to assess the pathological response of the primary tumor, comparing different antibody combinations with chemotherapy in potentially resectable patients. The long-term effect of initial and postoperative therapies was analyzed with recurrence-free survival (RFS) and overall survival (OS) as the endpoints. In this study, we were able to demonstrate a statistically significant better pathological response of the primary tumor and a significantly longer RFS for patients who received vascular endothelial growth factor (VEGF) antibody-based induction chemotherapy compared to patients who received epidermal growth factor receptor (EGFR) antibody-based therapy. These findings are clinically relevant for a prospective evaluation.

**Abstract:**

(1) Background: The pathological tumor response of the primary tumor to induction chemotherapy in synchronously metastasized colorectal cancer (mCRC) patients has not been investigated. The aim of this study was to compare patients treated with induction chemotherapy combined with vascular endothelial growth factor (VEGF) or with epidermal growth factor receptor (EGFR) antibodies. (2) Methods: We present a retrospective analysis, where we included 60 consecutive patients with potentially resectable synchronous mCRC who received induction chemotherapy combined with either VEGF or EGFR antibodies. The primary endpoint of this study was the regression of the primary tumor, which was assessed by the application of the histological regression score according to Rödel. The secondary endpoints were recurrence-free survival (RFS) and overall survival (OS). (3) Results: A significantly better pathological response and a longer RFS for patients treated with the VEGF antibody therapy compared to those treated with the EGFR antibodies was demonstrated (*p* = 0.005 for the primary tumor and log-rank = 0.047 for RFS). The overall survival did not differ. The trial was registered with clinicaltrial.gov, number NCT05172635. (4) Conclusion: Induction chemotherapy combined with a VEGF antibody revealed a better pathological response of the primary tumor, leading to a better RFS compared to that with EGFR therapy; this has clinical relevance in patients with potentially resectable synchronously mCRC.

## 1. Introduction

Colorectal cancer (CRC) is the second most common cause of cancer death worldwide [1]. Previous data describe up to 25% of these patients being diagnosed with synchronous liver metastases [2]; however, the actual incidence of patients presenting as synchronously metastasized is far larger, and a therapeutic approach for these patients is under discussion [3]. Due to new surgical techniques and chemotherapy combined with targeted therapies, the prognosis of metastatic colorectal cancer (mCRC) has markedly improved over the past few decades. Between 1999 and 2015, the 2-year relative survival rate for patients with stage IV disease increased from 21% to 35% [4]. Despite the still poor prognosis for patients with mCRC, surgery added to chemotherapy combined with antibody therapy seems to have improved patient outcomes. Several randomized clinical trials have highlighted the positive effect of induction chemotherapy with antibody therapy for patients with mCRC [5,6,7,8]. The FIRE-3 study demonstrated a better radiological response of the liver metastases to the epidermal growth factor receptor (EGFR) antibody cetuximab compared to the vascular endothelial growth factor (VEGF) antibody bevacizumab specifically in *RAS* wildtype patients, which translated into a significantly prolonged overall survival (OS) in the EGFR-treated patient group [5]. However, the American 80405 trial did not find any benefits in regard to response, recurrence-free survival (RFS), or OS when comparing these two strategies [9]. The addition of bevacizumab to the triplet FOLFOXIRI showed an impressive histological and radiological response and longer RFS and OS in several alternative studies [7,10,11,12,13,14].

In the recent randomized controlled FOxTROT trial, the feasibility, toxicity, morbidity, and efficacy of neoadjuvant chemotherapy for locally advanced colon cancer was investigated. In summary, the patients with locally advanced resectable colon cancer experienced a significant downstaging effect on the tumor without increased perioperative morbidity, although the EGFR antibody therapy, which was added for a proportion of the study population, did not add to the chemotherapy effect [15]. The multicenter randomized controlled phase II PRODIGE 22–ECKINOXE trial is currently investigating the benefit of neoadjuvant chemotherapy with cetuximab versus primary surgery for locally advanced colon cancers [16].

The histopathological response to systemic or radiation therapy is classified by grading systems. For primary CRC, several tumor regression scores have been established that indicate that a good histological response is a predictor of prolonged OS. In 2005, Rödel et al. implemented the regression score and showed a correlation between the regression score and RFS [17,18,19].

To our knowledge, there are no data concerning the histological response of the primary tumor in mCRC patients to neoadjuvant chemotherapy combined with antibody therapy. Starting induction therapy in mCRC patients with their primary tumor in situ bears the risk of local complications including obstruction, perforation, or bleeding; therefore, the response of the primary tumor and the metastatic sites needs to be understood. Very recent data from randomized trials evaluating the benefit of the resection of the primary tumor in unresectable mCRC demonstrated a worse outcome in the group, where the primary tumor was resected primarily [20,21].

In resectable mCRC patients, neither the treatment sequence nor the optimal therapy combination is standardized. The aim of this study was, therefore, to investigate and compare the histopathological response of the primary tumor to the addition of VEGF and EGFR antibodies to systemic therapy prior to the resection of the primary and liver metastases. The secondary endpoints of this study were RFS and overall survival (OS).

## 2. Materials and Methods

Our database was used to extract patients with synchronous mCRC who underwent induction chemotherapy with targeted agents prior to resection of their primary tumor and their metastases.

### 2.1. Patients and Therapy

We included patients with potentially resectable synchronously mCRC who underwent primary tumor and hepatic resection in two Viennese health network hospitals (Clinic Landstrasse (June 2014–March 2018) and Clinic Favoriten (April 2018–February 2021), which is the dedicated hepato-pancreato-biliary (HPB) center) between June 2014 and February 2021. Liver resections were performed by one dedicated HPB surgeon. All treatment decisions considering systemic chemotherapy and antibody therapy or surgical interventions were discussed by our local multidisciplinary tumor board (MDT). The resectability before and after induction chemotherapy was evaluated by a group of HPB surgeons using the FONG score [22].

We treated our patients with induction therapy combined with either a VEGF antibody or an EGFR antibody. The decision as to which antibody treatment was given depended on the patient’s comorbidities, *RAS* and *BRAF* status, and localization of the primary tumor [3]. After two months of therapy, a staging CT scan was initiated, and if the metastases were resectable, a liver-first strategy was indicated for patients requiring major liver surgery. The primary tumor was intended to be resected laparoscopically five weeks thereafter. If the liver only required minor resection, the primary tumor was resected synchronously. Adjuvant therapy was given to complete chemotherapy for a total treatment length of 6 months.

Every treatment decision concerning the induction therapy and whether a patient received an EGFR or a VEGF antibody, as well as the adjuvant therapy decision, were discussed by the MDT.

Patients with metachronous hepatic metastases were excluded. For the included patients, the following data were collected from their records: age, sex, type of liver resection (minor or major), location and type of primary resection, surgical treatment plan, type of targeted therapy prior to primary resection and liver resection, pathologic tumor response of the primary tumor according to Rödel et al. [19], *RAS* and *BRAF* status, tumor recurrence, location of tumor recurrence, 90-day morbidity and mortality, and overall survival status.

### 2.2. Pathological and Radiological Tumor Response Assessment

The pathological response of the primary tumor was assessed and described using the Rödel score by two independent pathologists who were blinded to the treatment received (Table 1) [19]. In our trial, we used the Rödel score for regression grading of the primary tumor because it is the standard regression score in our pathological department. The pathological response was assessed postoperatively in the surgical specimen 3 to 7 days after surgery. In order to ensure a uniform assessment, the pathologist in our HPB center (I.B.) reviewed the histological specimens from both treatment locations prior to these analyses.

The radiological tumor response of the liver metastases was assessed by one radiologist (C.V.) using pre- and post-treatment computer tomography (CT) or magnetic resonance imaging (MRI) of the liver with response evaluation criteria in solid tumors (RECIST, criteria 1.1) [23].

#### Ninety-Day Morbidity and Mortality

The postoperative complications after the primary tumor and liver resection were evaluated and described according to the classification system by Dindo et al. [24]. The Dindo classification is used to grade postoperative complications and divide them into five groups. Grade one contains any deviation from the normal postoperative course without needing any treatment. Grade two requires pharmacological treatment, and grade three requires surgical, endoscopic, or radiological intervention. Grade four describes life-threatening complications, and grade five results in death.

### 2.3. Follow-Up

After resection of both the primary tumor and metastases, follow-up investigations were performed every three months for the first two years. From year three to five, the time periods were extended to six months and to one year thereafter. Clinical examinations; contrast-enhanced CT scans of the thorax, abdomen, and pelvis; and blood tests including carcinoembryonic antigen (CEA) and cancer-antigen 19-9 (CA 19-9) were performed. Elevated tumor markers or suspect CT scans led to further investigations using MRI, contrast-enhanced sonography of the liver, or a positron emission tomography (PET)-CT scan.

### 2.4. Statistical Analysis

The primary endpoint was the pathological tumor response of the primary tumor. The secondary endpoints were RFS and OS. RFS was defined as the time from the last resection (liver or primary) until recurrence, death, or censored at the time of the last follow-up. OS was defined as the time from diagnosis until death or censored at the time of the last follow-up.

A descriptive analysis was performed for evaluation of the patients’ characteristics. The Mann–Whitney U test was used for comparative analysis. Continuous variables were described as the mean if normally distributed; otherwise, they were described as the median. Analyses of the RFS and OS were performed using Kaplan–Meier estimates and the log-rank test. *p*-values of <0.05 were considered significant. Statistical analyses were performed using SPSS version 23.0 (IBM, Armonk, New York, NY, USA).

## 3. Results

### 3.1. Patients

The database revealed 60 patients with synchronous mCRC, of whom 41 (68.3%) were male, and 19 (31.7%) were female. The median age was 62 (42–88) years. In 34 (56.7%) patients, the metastases were resected according to the liver-first strategy, whereas the primary tumor was resected first in 2 patients (3.3%), and the primary tumor and the liver metastases were resected simultaneously in 24 patients (40.0%). Concerning the primary tumor resection, there were 11 (18.3%) right-sided colonic resections, 24 (40.0%) left-sided resections, and 25 (41.7%) rectal tumor resections. Regarding the extent of the liver resections, 32 (53.3%) major and 28 (46.7%) minor resections were carried out (Table 2).

All 60 patients received induction chemotherapy with antibody therapy prior to resection of the primary tumor and the liver metastases. From these, 37 (61.7%) patients received a VEGF antibody, and 23 (38.3%) patients received an EGFR antibody. The VEGF antibody was bevacizumab, whereas the EGFR antibody was either cetuximab (*n* = 12, 52.2%) or panitumumab (*n* = 11, 47.8%). The type of the primary tumor, the surgical treatment of the primary tumor and the liver metastases, and the chemotherapy and antibody therapy are listed in Table 2.

In our study, there were two patients who required an acute resection of the primary tumor during induction therapy because of a progressive stenosis. One of them received a VEGF antibody and one an EGFR antibody.

Concerning the *BRAF* and the *RAS* status, we detected 3 (5.0%) patients with a *BRAF* mutation (mt) and 42 (70.0%) with the *BRAF* wildtype (wt), whereas in the remaining 15 (25.0%) patients, the *BRAF* status was unknown. In 22 (36.7%) patients, *RAS* mt tumors were detected, whereas 35 (58.3%) patients were wt and 3 (5.0%) patients had an unknown *RAS* status. The EGFR antibody treatment was only applied to patients with *RAS* wt status.

The chemotherapy comprised either capecitabine and oxaliplatin (CAPOX); 5-fluorouracil, leucovorin, and oxaliplatin (mFOLFOX6); 5-fluorouracil, leucovorin, and irinotecan (FOLFIRI); or 5-fluorouracil, leucovorin, oxaliplatin, and irinotecan (FOLFOXIRI).

Forty one (68.3%) patients received an adjuvant therapy, 16 (26.7%) did not receive any adjuvant therapy, and for 3 (5.0%), we could not assess the adjuvant therapy because the patients had their follow-up treatment at another hospital.

### 3.2. Radiological and Pathological Response

The comparison between the therapy with a VEGF and an EGFR antibody revealed a significant difference in the Rödel scores. Thirty-seven patients (61.6%) received chemotherapy combined with a VEGF antibody, and 23 (38.3%) received chemotherapy with an EGFR antibody (median Rödel scores for VEGF = 3 (CI 1.92–2.62); EGFR = 1 (CI 1.30–1.92); *p* = 0.005) (Table 3).

In Table 4, we demonstrate the association between the antibody treatment, the *RAS* status, and the Rödel score. In total, 22 patients with *RAS* mt received chemotherapy combined with VEGF antibody therapy and had a median Rödel score of three (CI 2–4). Another 22 patients with *RAS* wt received chemotherapy combined with EGFR antibody therapy and had a median Rödel score of one (CI 0–2). In addition, 13 patients with *RAS* wt received chemotherapy combined with VEGF antibody therapy and had a median Rödel score of three (CI 1–4). There was a statistically significant difference between the subgroups of patients with *RAS* mt who received a VEGF antibody therapy vs. patients with *RAS* wt who received an EGFR antibody therapy (*p* = 0.011), and there was also a statistically significant difference between the subgroups of patients with *RAS* wt who received a VEGF antibody therapy vs. patients with *RAS* wt who received an EGFR antibody therapy (*p* = 0.010).

The radiological response of the liver metastases showed an overall response rate (ORR) of 90.9% for the EGFR group and 78.9% for the VEGF group; therefore, a better radiological response for the EGFR antibody therapy was noted (Table 5). Two patients who were radiologically diagnosed with progressive disease (PD) intraoperatively showed a cystification of the metastases as a marker of response [25].

### 3.3. Ninety-Day Morbidity and Mortality

Regarding the 90-day morbidity and mortality after liver resection, we observed two (3.3%) patients with a Dindo I complication, six (10.0%) patients with a Dindo II complication, three (5.0%) with a Dindo IIIa complication, one (1.7%) with a Dindo IIIb complication, and one (1.7%) with a Dindo IVa complication. Three patients (5.0%) died; one after liver resection due to postoperative bleeding (*n* = 1), one with liver failure after liver partition with portal vein ligation for a staged hepatectomy (ALPPS procedure) (*n* = 1), and one with primary tumor anastomotic insufficiency after synchronous surgery (*n* = 1) (Table 2).

### 3.4. Survival Analyses

After a median follow-up of 21 months (0–80 months), 42 patients (70.0%) exhibited tumor recurrence. Of these, 19 (31.7%) suffered from intrahepatic tumor recurrence, nine (15.0%) patients from extrahepatic recurrence, and 14 (23.3%) patients from intra- and extrahepatic recurrence (Table 2). In patients who received induction chemotherapy with VEGF antibodies, the RFS was significantly longer than in patients who underwent induction chemotherapy combined with EGFR antibodies (*p* = 0.047, log-rank; VEGF median 12.33 (CI 10.39–14.26) months, EGFR median 8.40 (CI 5.85–10.95) months) (Figure 1 and Table 6). Regarding the OS, there was no significant difference between the two groups, although a numerical difference was noted (*p* = 0.066, log-rank; VEGF median 38.07 (CI 27.29–48.85) months, EGFR median 27.53 (CI 15.98–39.07) months) (Figure 2 and Table 6).

## 4. Discussion

In this retrospective study, we were able to demonstrate the benefit of induction chemotherapy combined with VEGF antibody therapy for patients with synchronous mCRC. We detected a better pathological response of the primary tumor (*p* = 0.005) and a better RFS (*p* = 0.047, log-rank) with the VEGF compared to the EGFR antibody therapy. Furthermore, we showed that the observed effect of the pathological tumor response was not associated with the *RAS* status (*p* = 0.011 and 0.010). The OS was numerically prolonged in patients receiving the VEGF combination without achieving statistical significance.

Over the past few decades, the treatment of synchronous mCRC has changed. Initially, it was assumed that patients with mCRC needed to undergo resection of the primary tumor to prevent obstructive or bleeding complications. In 2009, Poultsides at al. were the first to demonstrate that this may be unnecessary. They discovered that in 93% of the cases, palliative patients with mCRC who received combination chemotherapy did not require surgery for their primary tumor [26]. The international randomized controlled FOxTROT trial also demonstrated that neoadjuvant chemotherapy in patients with locally advanced resectable colon cancer had a significant downstaging effect on the tumor [27].

Therefore, we performed a retrospective study to investigate the tumor regression after induction chemotherapy combined with antibody therapy. Because of the clinical impression, where patients who received the VEGF antibody treatment experienced rapid symptom release and increased tumor regression, we evaluated the difference between the VEGF and EGFR antibody treatments.

Additionally, Ribero et al. discovered a reduction in the incidence and severity of hepatic injury for patients who received the VEGF antibody combined with induction chemotherapy [28]. Concerning the length of the induction therapy, the data have shown that extended induction chemotherapy does not improve the pathologic response, at least for liver metastases [29]. Furthermore, it is important to protect the liver from hepatic injury following long-term chemotherapy [16].

For the grading of the pathological response of CRC, there are multiple regression scores in use. In our study, the regression of CRC was graded with the Rödel score, and the pathologist was an important member of the MDT meeting since the description of the response to systemic therapy is valuable information for making an adjuvant treatment decision. The pathological response is routinely discussed after total neoadjuvant therapy (TNT) or radiotherapy (RT) alone [17,18,19], but should be included in every treatment decision of metastasized CRC patients in the future.

In examining the results of the radiological response of the liver metastases, we discovered a higher overall response rate for patients who received EGFR antibody treatment compared to the patients who received VEGF antibody treatment. Similar results have been reported, e.g., in the FIRE-3 study [5].

Synchronous mCRC has a poor prognosis with a two-year survival rate, reaching a maximum of 35% [4]. Therefore, it is important to standardize the therapy and treat the patients with the most efficient therapy options. A recent publication highlighted the fact that FOLFIRI-based chemotherapy had significant survival benefits compared to the FOLFOX regimen [30]. However, it is essential to take the patient’s profile into consideration prior to deciding the treatment plan, knowing that FOLFIRI-based chemotherapy can lead to increased morbidity. In particular, in the studied population it was found to be important to discuss additional treatment options with VEGF or EGFR antibody therapy to achieve a better response than with chemotherapy alone without increasing morbidity, which has to be investigated in the following prospective trials.

In the future, we are confident that we will be able to detect even more specific targeted therapies against cancer cells. A recently published paper indicated promising potential in the therapy regime of CRC using Affimer, a protein, which showed a higher drug accumulation in CRC cells using in vitro and in vivo models, and led to a significant decrease in tumor growth and an increased OS [31].

At this point, we believe that VEGF antibody therapy combined with induction chemotherapy seems to be the most effective way to increase tumor regression, relieve symptoms, and increase RFS in mCRC.

The *RAS* and *BRAF* status as well as the sidedness of the primary tumor could also have an impact on the efficacy of induction chemotherapy combined with antibody therapy and are important to include in the therapy decision made by the MDT. As our data demonstrated, VEGF antibody therapy had a better tumor response than EGFR antibody therapy independent of *RAS* status. Concerning the sidedness, Rossini et al. recently published a meta-analysis and highlighted the role of the primary tumor location in patients with *RAS* wt mCRC. They strongly recommended EGFR antibody therapy for patients with left-sided tumors and VEGF antibody therapy for patients with right-sided tumors [32].

We are aware that our work had major limitations. Due to the specific patient collective and the retrospective nature of the study, the sample size was quite low. Furthermore, the patients received different chemotherapy regimens (FOLFOX, FOLFIRI, and FOLFOXIRI), which could lead to a bias in our analyses. Therefore, our results need to be validated in upcoming prospective studies.

With our work, we could only show the trend that VEGF antibody therapy could be the better therapy option compared to EGFR therapy in patients with mCRC, and we believe that this approach for synchronous mCRC patients should further be explored in the future [8].

## 5. Conclusions

Our findings support the use of an induction treatment combination with VEGF antibodies for patients with potentially resectable synchronous mCRC, regardless of *RAS* status. The data revealed a statistically significantly better pathological tumor response of the primary tumor and improved RFS, which is consistent with our clinical impression of rapid symptom release and a prolonged midterm outcome.

## Figures and Tables

**Figure 1 cancers-15-02900-f001:**
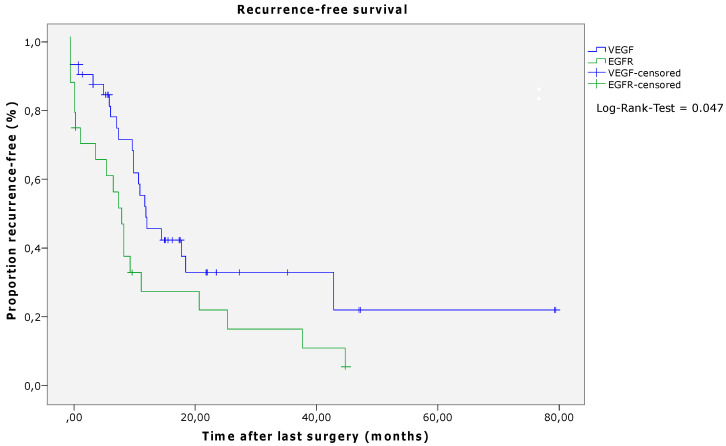
Recurrence-free survival according to antibody therapy (log-rank, *p* = 0.047). Abbreviations: VEGF—vascular endothelial growth factor; EGFR—epidermal growth factor receptor.

**Figure 2 cancers-15-02900-f002:**
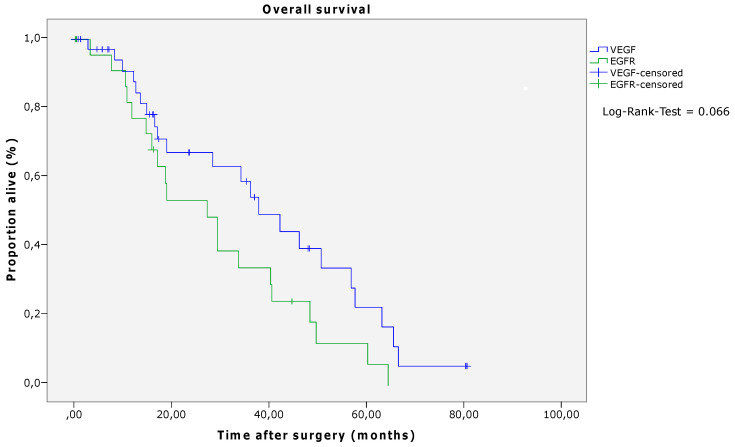
Overall survival according to antibody therapy (log-rank, *p* = 0.066). Abbreviations: VEGF—vascular endothelial growth factor; EGFR—epidermal growth factor receptor.

**Table 1 cancers-15-02900-t001:** TRG by Rödel et al. [19].

TRG	Tumor Regression
0	No regression
1	<25% of tumor mass
2	25–50% of tumor mass
3	>50% of tumor mass
4	Complete regression

Abbreviations: TRG—tumor regression grading.

**Table 2 cancers-15-02900-t002:** Patient characteristics (*n* = 60).

Patient Characteristics, *n*		60
Age, years, median (range)		62.0 (42–88)
Sex, *n* (%)	Male Female	41 (68.3)
		19 (31.7)
Liver resection, *n* (%)	Major Minor	32 (53.3) 28 (46.7)
Primary tumor, *n* (%)	Rectum Left colon and sigmoid colon Right colon	25 (41.7) 24 (40.0) 11 (18.3)
Surgical treatment, *n* (%)	Liver first Primum first Synchronous surgery	34 (56.7)
		2 (3.3)
		24 (40.0)
Induction chemotherapy with antibody prior to primary resection and resection of the liver metastases, *n* (%)	Chemotherapy + VEGF antibody therapy Chemotherapy + EGFR antibody therapy	37 (61.7)
		23 (38.3)
Tumor recurrence, *n* (%)	Yes No	42 (70.0)
		18 (30.0)
Location of tumor recurrence, *n* (%)	Intrahepatic Extrahepatic Intra- and extrahepatic	19 (31.7) 9 (15.0) 14 (23.3)
Ninety-day morbidity and mortality, Dindo et al. [24], *n* (%)	Dindo 0 Dindo I Dindo II Dindo IIIa Dindo IIIb Dindo IVa Dindo IVb Dindo V	45 (75.0)
		2 (3.3)
		6 (10.0)
		3 (5.0)
		1 (1.7)
		1 (1.7)
		0 (0)
		3 (5.0)

Abbreviations: VEGF—vascular endothelial growth factor; EGFR—epidermal growth factor receptor; *n*—number of patients.

**Table 3 cancers-15-02900-t003:** Comparison of the pathological tumor response of the primary tumor after induction chemotherapy with VEGF or EGFR antibody therapy.

Therapy	*n* = 60	%	Rödel Score, Median, 95% CI
Chemotherapy + VEGF antibody therapy	37	61.6	3 (2–4)
Chemotherapy + EGFR antibody therapy	23	38.3	1 (0–2)
*p*-value	0.005		

Abbreviations: VEGF, vascular endothelial growth factor; EGFR, epidermal growth factor receptor; CI, confidence interval.

**Table 4 cancers-15-02900-t004:** Comparison of the pathological tumor response of the primary tumor after induction chemotherapy with VEGF for *RAS* mt patients, EGFR antibody therapy for *RAS* wt patients, and VEGF for *RAS* wt patients.

Therapy	*n*	%	Rödel Score, Median, 95% CI
Chemotherapy + VEGF antibody therapy in patients with *RAS* mt	22	50.0	3 (2–4)
Chemotherapy + EGFR antibody therapy in patients with *RAS* wt	22	50.0	1 (0–2)
*p*-value	0.011		
Chemotherapy + VEGF antibody therapy in patients with *RAS* wt	13	37.1	3 (1–4)
Chemotherapy + EGFR antibody therapy in patients with *RAS* wt	22	62.9	1 (0–2)
*p*-value	0.010		

Abbreviations: VEGF—vascular endothelial growth factor; EGFR—epidermal growth factor receptor; *RAS*—rat sarcoma; mt—mutated; wt—wildtype; CI—confidence interval.

**Table 5 cancers-15-02900-t005:** The radiological response of the liver metastases after induction chemotherapy with the VEGF or EGFR antibody.

Antibody	ORR %	CR *n*, %	PR *n*, %	SD *n*, %	PD *n*, %
VEGF	78.9	0 (0.0)	30 (78.9)	6 (15.8)	2 (5.2)
EGFR	90.9	0 (0.0)	20 (90.9)	2 (10.0)	0 (0.0)

Abbreviations: VEGF—vascular endothelial growth factor; EGFR—epidermal growth factor receptor; ORR—overall response rate = complete response + partial response; CR—complete response; PR—partial response; SD—steady disease; PD—progressive disease.

**Table 6 cancers-15-02900-t006:** Recurrence-free survival and overall survival after induction chemotherapy with the VEGF or EGFR antibody (in months).

Antibody	RFS Median, 95% CI	OS Median, 95% CI
VEGF	12.33 (10.39–14.26)	38.07 (27.29–48.85)
EGFR	8.40 (5.85–10.95)	27.53 (15.98–39.07)

Abbreviations: RFS—recurrence-free survival; OS—overall survival; VEGF—vascular endothelial growth factor; EGFR—epidermal growth factor receptor; CI—confidence interval.

## Data Availability

The data presented in the study are available upon reasonable request from the corresponding author.

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
