# Peer review of "The Effect of Induction Chemotherapy with VEGF Inhibition on Tumor Response in Synchronously Metastasized Potentially Resectable Colorectal Cancer"

_cancers, 2023, doi:10.3390/cancers15112900_

Round 1

Reviewer 1 Report

The manuscript is clear and well written, but some methodological details
could be added 1. Justify the size of the sample considered, possibly also providing
an estimate of the power achievable by the study 2. More details about the possible confounding factors would be interesting,
analyzing the relative contribution (eg using a Cox model)
3. Confidence Intervals could be added to descriptive results

Author Response

The manuscript is clear and well written, but some methodological details 
could be added. 

Point 1: Justify the size of the sample considered, possibly also providing 
an estimate of the power achievable by the study 

Response 1: Although Clinic Favoriten is the dedicated Hepato-pankreato-biliary centre in Vienna and has consequently a high patient volume, unfortunately we didn’t have a larger sample size. Because of the specific patient collective and inclusion criteria, even though we used patient data from years 2014-2021, we were not able to collect a larger sample size. We added that factor to the limitations.

Point 2: More details about the possible confounding factors would be interesting, 
analyzing the relative contribution (eg using a Cox model).

Response 2: Thank you very much for your proposal, but unfortunately our statistical evaluation showed no statistical significant result for univariat and mulivariat cox regression, possibly due to the small sample size. This would be interesting to investigate in a larger population.

Point 3: Confidence Intervals could be added to descriptive results.

Response 3: Done.

Reviewer 2 Report

I would like to thank the Editor for allowing me to review this interesting manuscript. Thank you also to the Authors who shared their knowledge with the present study.

The Authors explored the effect of VEGF and EGFR inhibitors on the histopathologic response of the primary tumor. The target sample of their investigation was represented by patients suffering from potentially resectable metastatic colorectal cancer.

Below are my comments:

Major concerns:

1)   The main limitation of the present study is represented by its retrospective nature and its quite low sample size, especially considering the wide time span (2014-2021).

2)      Whether different first-line chemotherapeutic regimens may lead to different survival outcomes is still a matter of debate.

However, some recent papers (i.e. https://doi.org/10.3390/cancers14225513) highlighted how patients receiving FOLFIRI-based regimens had a significant survival advantage compared to those receiving FOLFOX. In the present study, the chemotherapeutic backbone encompassed 4 CT regimens for only 60 patients.

The Authors did not report any of these flaws as limitations of their study. I would recommend the Authors add a specific and detailed paragraph in the discussion section reporting the strengths but also the remarkable limitations of their work, rather than reporting again the treatment strategy adopted at their institution (already stated in the methods).

3)      A clear and detailed definition of potentially resectable disease is missing. I would suggest the Author specify which classification, scoring system, or criteria were adopted (i.e. Bittoni classification, Fong criteria…). Furthermore, whether or not each patient was judged upfront resectable, potentially resectable, or unresectable before and after induction systemic treatment by a group of specialized HPB surgeons should be specified.

Minor comments:

1)      Title: I think the title is a little bit misleading. Since the study focuses on the comparison between VEGF and EGFR inhibitors I would rephrase the title encompassing this concept.

2)      Abstract: I would suggest the Authors state more clearly also in the abstract that the target sample of the study is represented by potentially resectable mCRC cancer patients.

Introduction

3)      It is unclear to me the reason for reporting in the introduction a summary of the FOxTROT and PRODIGE 22 – ECKINOXE trials that are designed for patients suffering from locally advanced, resectable CRC rather than metastatic. Can the Authors please clarify this point?

4)      Lines 69 to 75: this paragraph should be moved to the methods section.

Methods

5)      The enrollment of patients suffering from mCRC with liver-only disease is kind of given for granted. I suggest specifying among the inclusion criteria that only patients without extrahepatic disease were deemed eligible for the purpose of the study.

6)      Lines 103-104: “All treatment decisions were discussed within our local multidisciplinary tumor board (MDT) prior to any systemic chemotherapy or a surgical intervention”. I would suggest the Authors clarify in more detail how their MDT works. Since 2 centers were involved (with the Clinic Favoriten stated as the dedicated HPB center), are all the cases of both centers discussed together, or 2 separate MDT exist?

7)      Lines 108-110: please specify and add the relative reference/es how you define major and minor liver resection.

Discussion and conclusion

8)      The discussion is quite synthetic and should be expanded, focusing on what may be the advantages of a more effective pathologic response of the primary tumor. Indeed, even though patients receiving anti-VEGF showed a higher median tumor regression grade compared to those receiving anti-EGFR, no clear advantage on OS was demonstrated. Furthermore, the RFS of patients receiving anti-VEGF was significant, but the result of the Log-rank test was very close to .05 (p = 0.047).

Moreover, in the presence of such a limited sample, the possibility of building a multivariate model in order to detect independent predictors of outcomes was precluded. That being said, I would suggest the Authors temper their statements, especially in the conclusions.

 An English language revision is recommended.

Author Response

I would like to thank the Editor for allowing me to review this interesting manuscript. Thank you also to the Authors who shared their knowledge with the present study. 

The Authors explored the effect of VEGF and EGFR inhibitors on the histopathologic response of the primary tumor. The target sample of their investigation was represented by patients suffering from potentially resectable metastatic colorectal cancer.

Below are my comments:

Major concerns:

Point 1: The main limitation of the present study is represented by its retrospective nature and its quite low sample size, especially considering the wide time span (2014-2021). 

Response 1: We recognize the main limitation by the retrospective nature of the study and the low sample size. Because of the inclusion criteria (synchronously metastasized, potentially resectable, colorectal cancer; liver first or synchronous resection, induction chemotherapy with VEGF or EGFR antibody, availability of Rödel score) we were not able to collect a larger sample size, even though we used patient data from years 2014-2021. Unfortunately we had to exclude several patients, who didn’t match the criteria. We mention it as a main limitation of the study.

Point 2: Whether different first-line chemotherapeutic regimens may lead to different survival outcomes is still a matter of debate.

However, some recent papers (i.e. https://doi.org/10.3390/cancers14225513) highlighted how patients receiving FOLFIRI-based regimens had a significant survival advantage compared to those receiving FOLFOX. In the present study, the chemotherapeutic backbone encompassed 4 CT regimens for only 60 patients.

The Authors did not report any of these flaws as limitations of their study. I would recommend the Authors add a specific and detailed paragraph in the discussion section reporting the strengths but also the remarkable limitations of their work, rather than reporting again the treatment strategy adopted at their institution (already stated in the methods).

Response 2: We have revised the discussion according to your recommendations.

Point 3: A clear and detailed definition of potentially resectable disease is missing. I would suggest the Author specify which classification, scoring system, or criteria were adopted (i.e. Bittoni classification, Fong criteria…). Furthermore, whether or not each patient was judged upfront resectable, potentially resectable, or unresectable before and after induction systemic treatment by a group of specialized HPB surgeons should be specified.

Response 3: Thank you for this proposal. We adapted a paragraph in the methods.

Point 4: Abstract: I would suggest the Authors state more clearly also in the abstract that the target sample of the study is represented by potentially resectable mCRC cancer patients.

Response 4: Done.

Minor comments:

Point 5: Title: I think the title is a little bit misleading. Since the study focuses on the comparison between VEGF and EGFR inhibitors I would rephrase the title encompassing this concept. 

Response 5: Done.

Reviewer 3 Report

This is a retrospective analysis of a small cohort of mCRC patients comparing the pathological response between patients receiving VEGF and EGFR ab based induction therapy. A tumour regression scale was used to measure outcome and both recurrence free survival (RFS) and overall survival (OS) were also analysed. Differences in regression and RFS was observed, and it is concluded that this has clinical relevance. I have highlighted the introduction and conclusions as needing specific attention, as it is the limited clarity of the rationale/experimental design, and the very general conclusions which makes no mention of RAS, that are of most concern.

The paper is reasonably well structured, and the data is presented transparently. However, the wording of key areas such as the abstract and conclusion make it difficult to establish the precise aim of the study. Is it a direct comparison between 2 treatment regimens? Is it testing one new treatment specifically (as suggested by the conclusion)? In addition, several variables within the cohort are not controlled which complicates the interpretation of the results, but these are not discussed/presented. As a result it is not obvious how clinically relevant the results actually are, and this needs to be elaborated upon.

Specifically:

1.     Data from two centres were used, but this is not mentioned further as a possible confounding factor. Can it be dismissed?

2.      Choice of treatment was decided by the clinician, but the criteria used are not discussed. Will this be potentially relevant to the RFS/OS results?

3.       One obvious element of the above is RAS mutation status. The EGFR group are RAS wild type, whereas 60% of the VEGF group have RAS mutations. Is the Rodel score associated with RAS status is the VEGF group? Again, does this need to be considered further - What do the results look like when the 12 RAS wild type tumours are removed from the VEGF group, so that it is a direct comparison of RAS wild type/EGFR v RAS mt/VEGF? Given the importance of this for treatment choice, this needs to be explicitly clear, and because the two are so interdependent, BOTH need to be covered in the discussion/conclusion sections.

4.       Is the type of liver resection (major/minor) split evenly between groups, and could this have an impact?

5.       The Title is not very informative. What is the take home message? What can the authors conclude from the work?

6.       The simple summary contains a fair amount of jargon, and the background section of the abstract does not explain what is being tested, only what is being analysed/measured.

While it is very difficult to control variables in a small retrospective study, the potential issues for interpretation do need to be mentioned in the discussion. The role of RAS in treatment choice, and the conflicting evidence regarding the impact of RAS mutation status on survival/prognosis, should be referred to. On line 268 it is suggested that practice has changed, to include VEGF – this is a bit cryptic. Does it mean all RAS mutated tumours were previously treated with EGFR, and this work is specifically testing the impact of this change? As per comments above, the rationale/design etc. needs to be laid out explicitly for the reader.

Minor textual points:

Lines 46, 240 – past few…?

Line 74 – DFS?

Lines 87, 92, 104 – prior to…?

None. Minor comments above. 

Author Response

This is a retrospective analysis of a small cohort of mCRC patients comparing the pathological response between patients receiving VEGF and EGFR ab based induction therapy. A tumour regression scale was used to measure outcome and both recurrence free survival (RFS) and overall survival (OS) were also analysed. Differences in regression and RFS was observed, and it is concluded that this has clinical relevance. I have highlighted the introduction and conclusions as needing specific attention, as it is the limited clarity of the rationale/experimental design, and the very general conclusions which makes no mention of RAS, that are of most concern.

The paper is reasonably well structured, and the data is presented transparently. However, the wording of key areas such as the abstract and conclusion make it difficult to establish the precise aim of the study. Is it a direct comparison between 2 treatment regimens? Is it testing one new treatment specifically (as suggested by the conclusion)? In addition, several variables within the cohort are not controlled which complicates the interpretation of the results, but these are not discussed/presented. As a result it is not obvious how clinically relevant the results actually are, and this needs to be elaborated upon.

Response: We have tried our best to revise the chapter.

Specifically:

Point 1: Data from two centres were used, but this is not mentioned further as a possible confounding factor. Can it be dismissed?

Response 1: It can be dismissed. Prof. Gruenberger moved in March 2018 from Klinik Lanstrasse to Klinik Favoriten. Confounding can be ruled out, because the lead surgeon, Prof. Gruenberger performed surgery in all included patients. The pathological and radiological reviews as well as the data collection were performed by one person each, retrospectively.

Point 2: Choice of treatment was decided by the clinician, but the criteria used are not discussed. Will this be potentially relevant to the RFS/OS results?

Response 2: Treatment decisions were discussed by several Hepato-pankreato-biliary surgeons, using the FONG-score, in our multidisciplinary tumor board, which consists of surgeons, onkologists, pathologists, radiologists and radiation therapists and therefore shouldn’t be relevant to the RFS/OS. The paragraph concerning the treatment decisions has been adapted.

Point 3: One obvious element of the above is RAS mutation status. The EGFR group are RAS wild type, whereas 60% of the VEGF group have RAS mutations. Is the Rodel score associated with RAS status is the VEGF group? Again, does this need to be considered further - What do the results look like when the 12 RAS wild type tumours are removed from the VEGF group, so that it is a direct comparison of RAS wild type/EGFR v RAS mt/VEGF? Given the importance of this for treatment choice, this needs to be explicitly clear, and because the two are so interdependent, BOTH need to be covered in the discussion/conclusion sections.

Response 3: Thank you very much for your proposal. We analysed the association of Rodel score between the groups of VEGF/RAS mt and EGFR/RAS wt and could show a statistically significant difference between the two groups. We added a new table (table 4) to the results and covered it in the discussion.

Point 4: Is the type of liver resection (major/minor) split evenly between groups, and could this have an impact?

Response 4: We listed the type of liver resections in the patient characteristicts. The type of liver resections is evenly distributed.

Point 5: The Title is not very informative. What is the take home message? What can the authors conclude from the work?

Response 5: Thank you. We changed the title according to your input.

Point 6: The simple summary contains a fair amount of jargon, and the background section of the abstract does not explain what is being tested, only what is being analysed/measured.

While it is very difficult to control variables in a small retrospective study, the potential issues for interpretation do need to be mentioned in the discussion. The role of RAS in treatment choice, and the conflicting evidence regarding the impact of RAS mutation status on survival/prognosis, should be referred to. On line 268 it is suggested that practice has changed, to include VEGF – this is a bit cryptic. Does it mean all RAS mutated tumours were previously treated with EGFR, and this work is specifically testing the impact of this change? As per comments above, the rationale/design etc. needs to be laid out explicitly for the reader.

Response 6: The abstract and the discussion have been modified and the impact of the RAS status has been pointed out as a limitation, because as you said, it is difficult to control variables in a small retrospective study. We are aware that there are major limitations because of the retrospective data collection and the small cohort, but therefore we just show a trend and want to initiate an upcoming prospective trial. But as we already mentioned in Response 3, we added a new table to the results, where you can see the comparison of a VEGF/RAS mt group vs a EGFR/RAS wt group.

Point 7: Minor textual points:

Lines 46, 240 – past few…?

Line 74 – DFS?

Lines 87, 92, 104 – prior to…?

Response 7: Thank you for the corrections. We adapted everything.

Reviewer 4 Report

This manuscript compared the pathological tumor response of the primary colorectal cancer (CRC) tumor to induction chemotherapy, combined with either anti-vascular endothelial growth factor (VEGF) or anti-epidermal growth factor receptor (EGFR) antibodies in synchronously metastasized CRC patients. The authors reported a significantly better pathological response (in terms of tumor regression) and longer regression free survival in patients treated with VEGF antibody therapy, although overall survival is not significantly different between the two arms of the trial.

Major comments for authors:

1.       In section 2.1, the research design should be more clearly described, preferably in a table or flow chart. E.g., indicate how the RAS and BRAF status, the chemotherapy regimen, tumor regression, 90-day morbidity and mortality, overall survival status of the 60 patients are stratified with respect to the treatment (anti-VEGF vs anti-EGFR). For instance, a recent publication (Eur J Cancer 2023 184 106) has reported that tumor side determines efficacy of antibody treatment for KRAS wildtype metastatic CRC. Do the authors observe this in this study?

2.       In Table 1, it would be clearer if the column labeled as ‘characteristics’ be relabeled as ‘Tumor Regression’.

3.       In the Discussion, there should be a paragraph describing the limitations of the study. E.g., the chemotherapy regimen for the 60 patients was not uniform; the results need to be validated in a prospective series since this is a retrospective study.

4.       The authors reported a better radiological response of the liver metastases for anti-EGFR antibody treatment. They should propose a reason for this observation (Table 4 and line 260 of Discussion).

5.       The authors should avoid using the term ’curative’ since the study is on synchronously metastatic CRC patients with no significant difference in overall survival between the two arms of the trial. E.g., on line 15 of the Simple Summary, the phrase ’curative treatable’ should be replaced with the phrase ‘resectable’. Similarly, in the Conclusion statement on line 284, the phrase ’after potentially curative treatment intention in’ should be replaced with the phrase ‘for potentially resectable’.

Minor Comments for authors:

1.       On line 118, the reference should be Table 1 (instead of Table 2).

2.       On line 129, the authors should give more details for the Dindo classification system.

3.       On line 210, the authors reported one (1.7%) patient has a Dindo IVa complication. However, this is not reflected in Table 2, which reported 0 patient in this category.

1.     The manuscript needs English editing. E.g., what does the term ‘progredient’ on line 171 mean? Further, the authors should avoid ending a sentence with just % (see e.g., line 245).

Author Response

This manuscript compared the pathological tumor response of the primary colorectal cancer (CRC) tumor to induction chemotherapy, combined with either anti-vascular endothelial growth factor (VEGF) or anti-epidermal growth factor receptor (EGFR) antibodies in synchronously metastasized CRC patients. The authors reported a significantly better pathological response (in terms of tumor regression) and longer regression free survival in patients treated with VEGF antibody therapy, although overall survival is not significantly different between the two arms of the trial. 

Major comments for authors:

Point 1: In section 2.1, the research design should be more clearly described, preferably in a table or flow chart. E.g., indicate how the RAS and BRAF status, the chemotherapy regimen, tumor regression, 90-day morbidity and mortality, overall survival status of the 60 patients are stratified with respect to the treatment (anti-VEGF vs anti-EGFR). For instance, a recent publication (Eur J Cancer 2023 184 106) has reported that tumor side determines efficacy of antibody treatment for KRAS wildtype metastatic CRC. Do the authors observe this in this study? 

Response 1: As a consequence of the low number of patients stratification does not seem appropriate to us. We mention this a an additional limitation. Thank you for pointing out this interesting publication. We added it to our discussion, because the sidedness is important for sure and should be investigated in following trials.

Point 2: In Table 1, it would be clearer if the column labeled as ‘characteristics’ be relabeled as ‘Tumor Regression’.

Response 2: Done.

Point 3: In the Discussion, there should be a paragraph describing the limitations of the study. E.g., the chemotherapy regimen for the 60 patients was not uniform; the results need to be validated in a prospective series since this is a retrospective study. 

Response 3: We have tried our best to revise the chapter.

Point 4: The authors reported a better radiological response of the liver metastases for anti-EGFR antibody treatment. They should propose a reason for this observation (Table 4 and line 260 of Discussion). 

Response 4: We added a section to the discussion.

Point 5: The authors should avoid using the term ’curative’ since the study is on synchronously metastatic CRC patients with no significant difference in overall survival between the two arms of the trial. E.g., on line 15 of the Simple Summary, the phrase ’curative treatable’ should be replaced with the phrase ‘resectable’. Similarly, in the Conclusion statement on line 284, the phrase ’after potentially curative treatment intention in’ should be replaced with the phrase ‘for potentially resectable’. 

Response 5: Done.

Minor Comments for authors:

Point 6: On line 118, the reference should be Table 1 (instead of Table 2).

Response 6: Thank you very much. That was our mistake.

Point 7: On line 129, the authors should give more details for the Dindo classification system. 

Response 7: Done.

Point 8: On line 210, the authors reported one (1.7%) patient has a Dindo IVa complication. However, this is not reflected in Table 2, which reported 0 patient in this category. 

Response 8: Thank you very much for your careful review. We corrected that.

Point 9: The manuscript needs English editing. E.g., what does the term ‘progredient’ on line 171 mean? Further, the authors should avoid ending a sentence with just % (see e.g., line 245).

Response 9: The paper has been checked by a native English speaker and corrections in syntax and grammar were made.

Reviewer 5 Report

The authors have done an excellent patient oriented study for comparing VEGF and EGFR based chemotherapy outcomes in colorectal cancers and its metastasis in liver. The article is indeed a good contribution in this field and could be accepted with some minor clarifications by the authors.

1. Although the authors conclude that VEGF based therapies have outperformed the EGFR in terms of patient pathological response. However in table 4, the ORR seems to be much higher in case of EGFR therapies. How do the authors justify this?

2. The group chosen in this study includes 41 males patients and 19 females (i.e  males ration double than female). Could it have any possibility that higher female numbers change the overall conclusion?

3. As the authors have investigated on EGFR and VEGF targeted therapies against colorectal cancer using antibodies in their work, in the discussion section they could also highlight the possibility of using other antibody like protein molecules (Affimers) which have potential for clinical trials against targeting the CEA antigen in colorectal cancer in the coming future (https://doi.org/10.1021/acsami.1c21655).  

4. In the Kaplan meier survival graph (figure 1 & 2), what does EGFR and VGFR censored mean?

Author Response

The authors have done an excellent patient oriented study for comparing VEGF and EGFR based chemotherapy outcomes in colorectal cancers and its metastasis in liver. The article is indeed a good contribution in this field and could be accepted with some minor clarifications by the authors.

Point 1: Although the authors conclude that VEGF based therapies have outperformed the EGFR in terms of patient pathological response. However in table 4, the ORR seems to be much higher in case of EGFR therapies. How do the authors justify this?

Response 1: We conclude that the VEGF based therapies have a better response on the primary tumor. The overall response rate characterises the radiolgical response of the liver metastases, which is consistent with current data.

Point 2: The group chosen in this study includes 41 males patients and 19 females (i.e  males ration double than female). Could it have any possibility that higher female numbers change the overall conclusion?

Response 2: To our knowledge there is no significant data considering different overall survival between men and women.

Point 3: As the authors have investigated on EGFR and VEGF targeted therapies against colorectal cancer using antibodies in their work, in the discussion section they could also highlight the possibility of using other antibody like protein molecules (Affimers) which have potential for clinical trials against targeting the CEA antigen in colorectal cancer in the coming future (https://doi.org/10.1021/acsami.1c21655).  

Response 3: Done. Thank you for the proposal.

Point 4: In the Kaplan Meier survival graph (figure 1 & 2), what does EGFR and VGFR censored mean?

Response 4: That means that the patients who didn’t have a recurrence or didn’t die at the end of the period of observation, were censored. EGFR censored were the patients, who received EGFR antibody treatment and were censored and VEGF censored where the patients, who received VEGF antibody treatment and where censored.

Round 2

Reviewer 2 Report

I want to thank the Authors for addressing some of my comments. Unfortunately, some of them have been ignored and are not even reported in the point-by-point response to reviewers. 

In my opinion, the manuscript has been moderately improved but some pitfalls still remain unsolved.

Before being suitable for publication I think the Authors should elaborate more on the discussion, which remains too synthetic for such an important topic. Furthermore, a professional English revision should be considered.

A professional English revision is required

Author Response

I want to thank the Authors for addressing some of my comments. Unfortunately, some of them have been ignored and are not even reported in the point-by-point response to reviewers. 

In my opinion, the manuscript has been moderately improved but some pitfalls still remain unsolved.

Before being suitable for publication I think the Authors should elaborate more on the discussion, which remains too synthetic for such an important topic. Furthermore, a professional English revision should be considered.

Comments on the Quality of English Language: A professional English revision is required

Response: We are very sorry, you have the impression, we ignored some of your comments. We tried to respond to your previous comments again and tried our best to elaborate the discussion.

The manuscript was sent to the cancers language editing service and revised by a professional english editor and changes in syntax and grammar have been made.

Reviewer 3 Report

The changes made have improved the clarity of the manuscript, and the limitations of the study are now acknowledged in the discussion. I do have several remaining comments, the most important relating to the new table which has been added, and how it is interpreted:

Additional detail concerning therapeutic decisions is useful, but the main factors which result in patients being placed on one treatment or the other are not outlined. Can these be briefly mentioned or an appropriate citation given?  

Response to point 1. That a single surgeon was involved could be mentioned in the text.

The additional information relating to RAS in Table 4 is welcome, although what is currently presented doesn’t show the response is stronger in RAS mt patients, as stated on lines 347-348. The thrust of my original comment related to RAS as a potential confounding factor – “Is the Rodel score associated with RAS status in the VEGF group?” The comparison shown is secondary as the two groups included are different with respect to both antibody used and RAS status. For this reason, inclusion of the means/stats for the VEGF-RAS wt group in Table 4 is needed, if this table is to be included. While there will be fewer patients in the 3rd group, it would provide transparency in terms of the relationship between RAS status and the antibody used, and may make it clear that the observed effect on score is unlikely to be due to RAS status. It could even presented as Table 3B.

The CI values for the Rodel scores are now rounded to whole numbers in the table, but given to 2 decimal places elsewhere. Please make these consistent, or explain why both are used. 

Minor errors in language and punctuation will need to be corrected. The revised discussion would benefit from the merging of paragraphs as appropriate, as some currently resemble bullet points.

Author Response

The changes made have improved the clarity of the manuscript, and the limitations of the study are now acknowledged in the discussion. I do have several remaining comments, the most important relating to the new table which has been added, and how it is interpreted:

Additional detail concerning therapeutic decisions is useful, but the main factors which result in patients being placed on one treatment or the other are not outlined. Can these be briefly mentioned or an appropriate citation given? 

Response: We added a new paragraph according to your comment. (Line 107-108)

Response to point 1. That a single surgeon was involved could be mentioned in the text. 

Response: Thank you. We added that to the text. (Line 101)

The additional information relating to RAS in Table 4 is welcome, although what is currently presented doesn’t show the response is stronger in RAS mt patients, as stated on lines 347-348. The thrust of my original comment related to RAS as a potential confounding factor – “Is the Rodel score associated with RAS status in the VEGF group?” The comparison shown is secondary as the two groups included are different with respect to both antibody used and RAS status. For this reason, inclusion of the means/stats for the VEGF-RAS wt group in Table 4 is needed, if this table is to be included. While there will be fewer patients in the 3rd group, it would provide transparency in terms of the relationship between RAS status and the antibody used, and may make it clear that the observed effect on score is unlikely to be due to RAS status. It could even presented as Table 3B.

Response: Thank you for the explanation. We appreciate your idea and changed the Table and the paragraph in the results (Line 220-232) and the discussion (Line 283-284).

The CI values for the Rodel scores are now rounded to whole numbers in the table, but given to 2 decimal places elsewhere. Please make these consistent, or explain why both are used. 

Response: The CI values for the Rodel score don’t have 2 decimal places, because the Rodel score, as you can see in Table 1 is a score, which only has integers. Therefore we find it more rational to round the CI to whole numbers.

Comments on the Quality of English Language

Minor errors in language and punctuation will need to be corrected.

Response: The manuscript was sent to the cancers language editing service and revised by a professional english editor and changes in syntax and grammar have been made.

The revised discussion would benefit from the merging of paragraphs as appropriate, as some currently resemble bullet points.

Response: Thank you, we tried to adapt the discussion according to your comment.

Reviewer 4 Report

The revised manuscript (v2) has addressed most of the reviewers’ concerns satisfactorily.

Further comments for authors:

1.       Since the manuscript compares two antibody therapies, a more suitable title maybe: Effect of antibody-cum-induction chemotherapy on tumor response in synchronously metastasized potentially resectable colorectal cancer.

2.       The statistical analysis for the significant result comparing VEGF/RAS mt and EGFR/RAS wt treatment should be reported in Table 4 before the p-value.

3.       There are still some English language problems detected. On line 431, the correct statement should be “… a retrospective study”. On line 434, it should be: ‘… important to include in the therapy decision at the MDT”.  On line 445, it should be: “… and we believe that this approach …”. Missing/more appropriate word is underlined.

There are still some English language problems detected. On line 431, the correct statement should be “… a retrospective study”. On line 434, it should be: ‘… important to include in the therapy decision at the MDT”.  On line 445, it should be: “… and we believe that this approach …”. Missing/more appropriate word is underlined.

Author Response

The revised manuscript (v2) has addressed most of the reviewers’ concerns satisfactorily. 

Further comments for authors:

Since the manuscript compares two antibody therapies, a more suitable title maybe: Effect of antibody-cum-induction chemotherapy on tumor response in synchronously metastasized potentially resectable colorectal cancer. 

Response 1: Thank you for your idea, but we consider the adapted title appropriate.

The statistical analysis for the significant result comparing VEGF/RAS mt and EGFR/RAS wt treatment should be reported in Table 4 before the p-value. 

Response 2: Thank you. We corrected that.

There are still some English language problems detected. On line 431, the correct statement should be “… a retrospective study”. On line 434, it should be: ‘… important to include in the therapy decision at the MDT”.  On line 445, it should be: “… and we believe that this approach …”. Missing/more appropriate word is underlined.

Response 3: Thank you very much for your suggestions for improvement. We adapted the paragraphs according to your comments. The manuscript was sent to the cancers language editing service and revised by a professional english editor and changes in syntax and grammar have been made.

Round 3

Reviewer 2 Report

I would like to thank the Authors for providing the revised version of the manuscript. After the English language revision, and merging the replies to all reviewers, I feel the paper has significantly improved.

Thank you for replying to my previous unanswered comments, and for providing a more comprehensive discussion section. I think the manuscript is now suitable for publication.

Congratulations.